# *Djnedd4L* Is Required for Head Regeneration by Regulating Stem Cell Maintenance in Planarians

**DOI:** 10.3390/ijms222111707

**Published:** 2021-10-28

**Authors:** Qingnan Tian, Yujia Sun, Tingting Gao, Jiaxin Li, Huimin Fang, Shoutao Zhang

**Affiliations:** 1School of Life Sciences, Zhengzhou University, Zhengzhou 450001, China; tianqn@zzu.edu.cn (Q.T.); yujiasun903@163.com (Y.S.); gaotingting406@163.com (T.G.); ljx15203995016@163.com (J.L.); 2Henan Key Laboratory of Bioactive Macromolecules, Zhengzhou 450001, China

**Keywords:** *Djnedd4L*, *Djubc9*, stem cells, regeneration, SUMOylation, ubiquitylation

## Abstract

SUMOylation and ubiquitylation are homologous processes catalyzed by homologous enzymes, and they are involved in nearly all aspects of eukaryotic biology. Planarians, which have the remarkable ability to regenerate their central nervous system (CNS), provide an excellent opportunity to investigate the molecular processes of CNS regeneration in vivo. In this study, we analyzed gene expression profiles during head regeneration with an RNA-seq-based screening approach and found that *Djnedd4L* and *Djubc9* were required for head regeneration in planarians. RNA interference targeting of *Djubc9* caused the phospho-H3 mitotic cells to decrease in quantity, or even become absent as a part of the *Djubc9* RNAi phenotype, which also showed the collapse of the stem cell lineage along with the reduced expression of epidermal differentiation markers. Furthermore, we found that *Djnedd4L* RNAi induced increased cell division and promoted the premature differentiation during regeneration. Taken together, our findings show that *Djubc9* and *Djnedd4L* are required for stem cell maintenance in the planarian *Dugesia japonica*, which helps to elucidate the role of SUMOylation and ubiquitylation in regulating the regeneration process.

## 1. Background

Protein ubiquitination is a multifaceted posttranslational modification that can lead to protein degradation by the 26S proteasome [1]. Protein ubiquitination is attached to substrates by a mechanism that involves a number of factors, beginning with an E1 ubiquitin-activating enzyme that makes ubiquitin reactive. Activated ubiquitin is conjugated to an E2 ubiquitin conjugase, which interacts with an E3 ubiquitin ligase to transfer ubiquitin onto a target substrate [2,3,4]. The process of SUMOylation is similar to ubiquitylation, with small ubiquitin-like modifier (SUMO) attaching to substrates through an E1-activating enzyme, an E2-conjugating enzyme, and an E3-ligase [5]. Ubiquitination and SUMOylation covalently modify various target proteins in all eukaryotes. Depending on the targets, SUMOylation and ubiquitylation regulate various cellular mechanisms, such as protein–protein interactions, protein stability, subcellular localization, cell cycle regulation, and transcription factor activity [4,6,7,8,9].

Freshwater planarian flatworms are among the few animals with the remarkable ability to regenerate any part of their body, including the central nervous system (CNS) from tissue fragments originating from almost any part of their bodies [10,11,12,13,14]. These properties require a pool of adult stem cells named neoblasts [15,16]. Neoblasts can migrate toward the wound site after injury and differentiate into any planarian cell type to regenerate the missing target tissues. Thus, in the process of regeneration, neoblasts must receive proper instructions to correctly achieve the replacement of missing tissues [14,17,18,19,20]. However, how ubiquitylation participates in planarian regeneration has not been thoroughly elucidated to date.

In this study, we characterize the planarian homologue of *Djnedd4L*, a member of E3 ubiquitin ligases, and *Djubc9*, a sole E2-conjugating enzyme gene, in *Dugesia japonica*. RNAi knockdown of *Djubc9* leads to a lysis phenotype in both regenerating and intact planarians. We observed a depletion of cell division along with a loss of stem cell progeny markers. RNAi knockdown of *Djneddll4* results in the absence of the anterior marker and the reduced expression levels of the head makers. Our results suggest that *Djubc9* is required for the maintenance of stem cells in planarians and *Djnedd4L* is required for the reestablishment of anterior domain identities during planarian regeneration.

## 2. Results

### 2.1. Transcriptome Analysis Clarified the Genes Involved in Head Regeneration in the Planarian Dugesia japonica

To obtain the gene expression profiles during planarian head regeneration, we analyzed the transcriptomes of planarians 3 days after head amputation (Figure 1A). The anterior pre-pharyngeal fragments frozen immediately after decapitation were used as controls (0 dpa). Using the assembly genome of *Dugesia japonica* as reference (BioProject accession: PRJNA580305), a total of 23497 genes were generated by sequencing on an Illumina Genome Analyzer II, and 10576 genes were annotated in the MAKER genome annotation. By comparing the transcriptomes of head regeneration and control tissues, we identified 763 downregulated transcripts (fold change < 0.5) and 379 upregulated transcripts (fold change > 2) (Figure 1B,C). The Kyoto Encyclopedia of Genes and Genomes (KEGG) functions analysis of the differentially expressed genes (DEGs) revealed that “Metabolic pathways” had the highest number of genes (Figure 1D). Interestingly, we found that the genes coding ubiquitination and SUMOylation enzymes were DEGs during head regeneration (Appendix A).

### 2.2. Djnedd4L and Djubc9 RNAi Cause Head Regeneration Defects

Next, we performed an RNAi screen to identify the genes need for planarian head regeneration and found that *Djnedd4L* and *Djubc9* were required for planarian regeneration. After 8 days of regeneration, the control animals had a well-formed head and tail (10/10) and a pair of eyes that could be observed clearly. In contrast, following *Djubc9* RNAi, the anterior trunk and tail fragments formed a significantly reduced blastema and lacked eyes (20/20) (Figure 2A). In addition to the reduced blastema, the expression of synapsin, a CNS marker [21,22,23], showed that although the CNS regenerated a complete nerve cord, the cephalic brain did not form in the “anterior blastema” of the *ubc9* RNAi trunk and tail fragments (8/10) (Figure 2D). These results show that with *Djubc9* (RNAi), the blastemas could not appropriately differentiate into the anterior region. In *Djnedd4L* RNAi-treated animals, the eyes could barely be observed and the triangular head could not form (Figure 2D). The CNS could not regenerate completely in the newly formed head in the trunk and tail fragments (Figure 2D). 

Whole-mount RNA in situ hybridization (WISH) was performed in intact and regenerative worms to analyze the expression patterns of *Djnedd4L* and *Djubc9* in planarians. Worms were amputated pre- and post-pharyngeally and fixed at 1 and 3 d post amputation. We observed a ubiquitous expression throughout the body with no specific localization to tissue or organs in intact animals (Figure 2B). The expression level of *Djubc9* was higher in the wound region during regeneration (Figure 2B), which is consistent with the transcriptomes of head regeneration (Appendix A). Similarly, the intact animals stained for *Djnedd4L* also showed ubiquitous expression consistent with the expression patterns of *Djubc9* in intact and regenerating animals (Figure 2C). These data raise the possibility that *Djubc9* and *Djnedd4L* might be required not only for the regeneration process. Next, we tested whether *Djnedd4L* and *Djubc9* were also required for tissue homeostasis (Figure 2E). We found that only *Djubc9* (RNAi) has an effect on the tissue homeostasis. At day 10, the border of the head also began to curl and lyse (18/20) in *Djubc9* (RNAi) animals. At day 16, the lysis region extended to the whole body (17/20) (Figure 2E). We propose that knockdown of *Djubc9* or *Djnedd4L* may disrupt stem cell maintenance during tissue regeneration. Therefore, we next analyzed the effect of *Djubc9* and *Djnedd4L* on cell division and the maintenance of stem cells during the course of the *cdc42* RNAi and *Djnedd4L* RNAi phenotypes. 

### 2.3. Djubc9 (RNAi) Phenotype Defects Indicate That Djubc9 Is Required for the Maintenance of Stem Cells

To assess whether *Djubc9* is required for cell division, we performed an analysis of the mitotic response by quantifying pH 3-positive cells in intact animals. The pH 3-positive cells in *Djubc9* RNAi animals were comparable with those of the controls in the first three days. After 5 days of RNAi, a progressive reduction in pH 3-positive cells in intact animals could be detected (Figure 3A). These results suggest the *Djubc9* RNAi might disrupt the maintenance of stem cell progeny. Therefore, we next examined the stem cell lineage makers in *Djubc9* RNAi animals. We used three identified lineage markers of planarian stem cells and two differentiated cell markers to identify the cell types that were affected by hypo-proliferation, including stem cell (*Djpiwi-1*), early progeny (*prog*), late progeny (*AGAT1*), later stage (*Vimentin*), and mature cell (*Lamin B*) [24,25,26,27,28,29,30,31,32,33]. We examined the expression of lineage production for the *Dj**ubc9* phenotype. A significant reduction in the expression of stem cell-specific markers was detected. The expression of *Djpiwi-A*, as well as of *prog,* and *AGAT1*, was nearly absent (Figure 3B–D). The expression levels of two differentiated cell markers (*Vimentin* and *Lamin B*) for the epidermal lineage were also decreased (Figure 3E,F). These results suggest that *Djub**c9* is needed for stem cells and progeny, as *Djubc9 RNAi* animals lost their stem cells and progeny. 

### 2.4. The Effect Djnedd4L (RNAi) on Cell Division and Differentiation during Regeneration 

The expression of the CNS marker *PC-2* showed that the *Djnedd4L* RNAi animals formed a blastema without a brain (Figure 4A). To further evaluate whether *Djnedd4L* RNAi only affects head regeneration, we performed in situ hybridization to detect the head and tail markers, secreted frizzled protein 1 (*sFRP1*) and frizzled-like protein T (*fzT*) in trunk fragments [34]. Eight days after amputation, the control animals showed normal *sFRP1* expression patterns in the anterior head regions. In contrast, the *Djnedd4L* RNAi animals almost lost the expression of *sFRP1* (Figure 4B). Similarly, the tail marker *fzT* also showed decreased expression compared to the controls (Figure 4C). Next, we used qRT-PCR to analyze the effect of *Djnedd4L* on the expression levels of multiple position control genes (PCGs) during regeneration of trunk fragments after *Djnedd4L* RNAi. Total RNA used in qRT-PCR was isolated from trunk fragments; *sFRP-1*, *chat*, *otxa*, and *notum* were selected as head PCGs, and *AxinB*, *wntP*, *fzt*, and *Djabdba* were selected as tail PCGs [14,35,36,37,38,39,40,41]. After 4 days of RNAi, the expression levels of the head PCGs were reduced in the *Djnedd4L* RNAi animals compared to the controls. Similarly, the expression levels of the tail PCGs were also decreased (Figure 5D). These results further indicate that *Djnedd4L* is required not only for head regeneration but also tail regeneration.

As stem cell function is essential to planarian regeneration, it was possible that *Djnedd4L* (RNAi) disrupted planarian stem cell lineages. Thus, we first evaluated mitotic activity by whole-mount immunofluorescence for phosphorylated histone H3 (H3ser10p), a marker of dividing cells beginning at the G2 /M cell phase, in *Djnedd4L* (RNAi) regenerating trunks 8 days post-amputation. We found that the number of phospho-H3+ cells in *Djnedd4L* (RNAi) animals were significantly increased compared to the controls (Figure 5A). Correspondingly, the expression levels of piwi-1-marked undifferentiated neoblasts were also increased after *Djnedd4L* RNAi (Figure 5B). Consistent with this, the expression patterns of prog and agat, which labeled the early progeny and late progeny, were increased after *Djnedd4L* RNAi (Figure 5C,D). Similarly, the expression of vim in differentiated epidermal cells not integrated into the epidermis was also significantly increased (Figure 5E). However, the marker for mature epidermal cells, lamin B, expression in *Djnedd4L* (RNAi) animals was significantly decreased (Figure 5F). Taken together, our findings suggest that *Djnedd4L* RNAi induced increased cell division and promoted the premature differentiation during regeneration. These results further indicate that *Djnedd4L* participates in tissue regeneration by regulating stem cell proliferation and differentiation. 

## 3. Discussion 

Neurodegenerative diseases can cause irreversible damage to the CNS. Therefore, it is crucial to understand the basic molecular mechanisms needed to induce and promote the reestablishment of nervous system function. Planarians, which have a remarkable ability to regenerate their CNS, provide an excellent opportunity to analyze this process in vivo. To shed light onto the control of CNS regeneration, we analyzed the gene expression profiles with an RNA-seq-based screening approach and found that *Djnedd4L* and *Djubc9* were required for CNS regeneration in planarians. 

The ubiquitin system is an essential cellular pathway in protein regulation. In another other model planarian species, *Schmidtea mediterranea*, *Smedubc9* has been proven to control stem cell proliferation and regional cell death [42]. Consistent with the studies in *Schmidtea mediterranea*, the pH 3-positive cells in the *Djubc9* RNAi animals were nearly absent in both regenerated and intact planarians (Figure 3A and Appendix A), in which the stem cell lineage collapsed along with the reduced expression of epidermal differentiation markers (Figure 3B–F). These data suggest that SUMOylation is required for the maintenance of stem cells in both *Schmidtea mediterranea* and *Dugesia japonica*.

In previous reports, specific roles for ubiquitin ligases have been characterized in planarian regeneration. Some E3 ubiquitin ligases, such as huwe1, wwp1, trip12, and Cullin-RING ubiquitin ligase (CRL) complexes, have been proven to be required for tissue regeneration by regulating the neoblast population and tissue specification [43,44]. In this study, we characterized the role of the HECT-domain E3 ligase, nedd4L, in tissue renewal. We found that the expression of the head marker *sFRP-1* was absent in the *Djnedd4L* RNAi animals in the regenerating animals. The expression levels of head and tail markers were both reduced in the intact *Djnedd4L* RNAi animals, suggesting that *Djnedd4L* is required for not only head regeneration but also tail regeneration. In addition to this, similar to the phenotypes observed after knockdown of HUWE1 in *S. mediterranea*, *Djnedd4L* RNAi induced hyper-proliferation and abnormal differentiation of stem cells were observed, suggesting that it is a regulator of cell cycle. Consistent with this, nedd4L has been reported as a tumor suppressor by regulating several protein targets, such as LGR5 and DVL2 [45,46]. Thus, our studies provide an excellent model to investigate the mechanisms of nedd4L regulating the cell cycle during stem cell self-renewal and differentiation. 

Planarians are an excellent model system for experimental dissection of processes, such as tissue regeneration and tissue patterning. Planarian regeneration requires neoblasts to produces new cells of blastemas, which generate the replacement parts needed for regeneration. The fate specification in planarian regeneration primarily occurs in neoblasts, which are the source of new cells in planarian regeneration to produce all adult cell types. The neoblast’s capacity for renewal and differentiation suggests that the neoblast population contains stem cells [47]. During the processes of homeostasis and regeneration, stem cells must not only continually maintain the rates of cell division but also properly induce differentiated progeny into new tissues. Our findings show that SUMOylation and ubiquitylation are required for regeneration by regulating the maintenance of stem cell in the planarian *Dugesia japonic**a*, which helps to elucidate the role of SUMOylation and ubiquitylation in regulating the regeneration process.

## 4. Materials and Methods

### 4.1. Planarian Culture

All experiments were performed with a clonal strain of the planarian *Dugesia japonica*. Planarians were maintained as previously described [48]. The animals were starved for at least 7 days prior to the experiments.

### 4.2. cDNA Clones and RNAi Experiments

*Djubc9* and *Djnedd4L* were amplified from cDNA in a constructed library of expressed sequence tags (ESTs) (Appendix A). dsRNA was synthesized as previously described [49,50]. In the case of single RNAi experiments, 2 μg/μL of dsRNA was injected into the animals using microinjection twice a day for three days. dsRNA against the green fluorescent protein (GFP) sequence was injected into the control animals. The animals were transversely cut into three fragments anterior and posterior to the pharynx and allowed to regenerate for the indicated number of days. The primers for cloning and dsRNA generation are listed in Appendix A. 

### 4.3. In Situ Hybridization

Whole-mount in situ hybridizations (WISHs) were performed as previously described with digoxigenin-labeled probes [51]. The samples were hybridized with a DIG-labeled probe at 56 °C for 17 h. Subsequently, colorimetry (NBT/BCIP) was used to detect the signal. 

### 4.4. Whole-Mount Immunostaining

Whole-mount immunostaining was performed as previously described [52]. The animals were killed with 5% NAC in phosphate-buffered saline (PBS) for 5 min at room temperature and washed four times with PBS containing 0.1% TritonX-100 (PBST). The animals were then fixed in PBST containing 4% paraformaldehyde, and cold 100% MeOH for 1 h. Next, animals were blocked with 10% goat serum in PBST for 2 to 4 h at 4 °C and incubated with primary anti-synapsin (1:100–1:500 Developmental Studies Hybridoma Bank) or anti-H3P (1:500; Millipore, Burlington, MA, USA, 05-817R) antibodies overnight. The secondary antibodies include goat anti-rabbit Alexa Fluor 568 (1:500; Invitrogen, Waltham, MA, USA, 11036) for anti-H3P, and goat anti-mouse Alexa Fluor 488 (1:500; Invitrogen, 673781) for anti-synapsin. Digital pictures were collected using NIS element software (Nikon, Tokyo, Japan).

### 4.5. Quantitative Real-Time PCR Analysis of Gene Expression

Quantitative real-time PCR was performed as previously described [49]. Total RNA was extracted from each pool of planarians using Trizol (TaKaRa, Gunma, Japan), and cDNA was generated from 1 μg of total RNA with oligo-dT primers and reverse transcriptase (TaKaRa). Three samples were run in parallel for each condition and normalized to the expression of *elongation factor 2* (*EF-2)*. The primers used for quantitative real-time PCR are listed in Appendix A. 

### 4.6. Sample Collection and Head Regeneration RNA-Seq

Regeneration was allowed to occur for 3 days for the preparation of the head regeneration sequencing libraries for RNA-Seq. Fragments were obtained by a second cut anterior to the pharynx. The anterior pre-pharyngeal fragments frozen immediately after decapitation were used as controls. Ten planarian prepharyngeal fragments were pooled. The libraries were sequenced on an Illumina Genome Analyzer II as previously described [53].

## Figures and Tables

**Figure 1 ijms-22-11707-f001:**
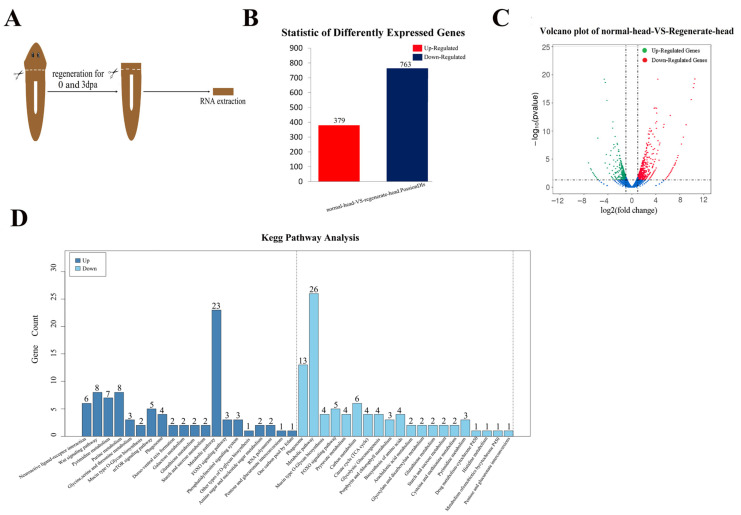
The head regeneration transcriptome of the planarian *Dugesia japonica*. (**A**) Schematic overview of the RNA sequencing approach. (**B**,**C**) Differentially expressed genes detection (DEGs). (**B**) Summary of DEGs. The X axis represents comparison samples. The Y axis represents DEG numbers. (**C**) Volcano plot of DEGs. The Y axis represents −log10 transformed significance. The X axis represents log2 transformed fold change. (**D**) The Kyoto Encyclopedia of Genes and Genomes (KEGG) function annotation of DEGs.

**Figure 2 ijms-22-11707-f002:**
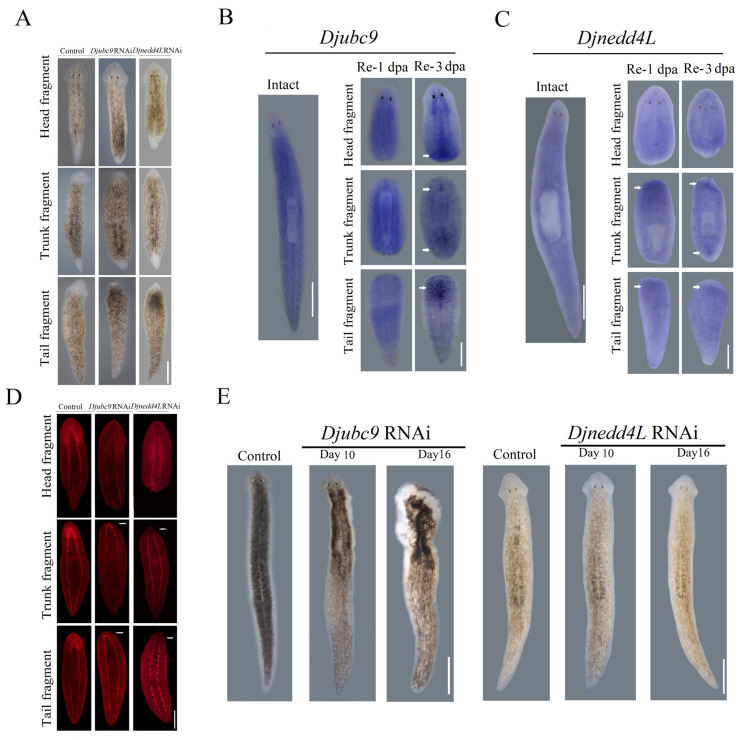
Effects of *Djubc9* RNAi and *Djnedd4L* RNAi during regeneration. (**A**) Regeneration defects caused by *Djubc9* RNAi and *Djnedd4L* RNAi. (**B**,**C**) Expression patterns of *Djubc9* (**B**) and *Djnedd4L* (**C**) in intact and regenerating animals (n = 20) assessed by whole-mount in situ hybridization. The worms were fixed 1 and 3 d after amputation. The arrowheads point to the higher expression of *Djnedd4L* and *Djubc9* in wound region. (**D**) The CNS defects caused by *Djubc9* RNAi and *Djnedd4L* RNAi. The whole-mount immunostainings were performed with primary anti-synapsin. The arrows point to under-developed brains and to a blastema without a brain. (**E**) Intact phenotypes caused *Djubc9* RNAi and *Djnedd4L* RNAi. A stereotypical lysis phenotype beginning around day 10 in *Djubc9* RNAi animals, which proceeded until worms completely curled and lysed at about day 16. Scale bars: 300 µm.

**Figure 3 ijms-22-11707-f003:**
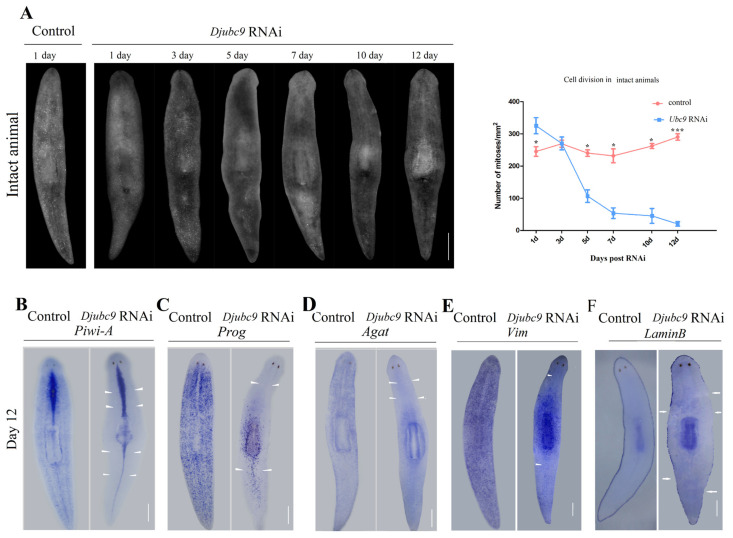
Analysis of the effect of *Djubc9* RNAi on cell division and stem cell lineage. (**A**) Analysis of the effect of *Djubc9* RNAi on cell division. Intact worms were stained following RNAi injections using the marker H3ser10p, which marks cells during the G2/M cell cycle transition. (**B**–**F**) The expression patterns of epidermal lineage markers of stem cells (*piwi*) (**B**), early progeny (*prog*) (**C**), late progeny (*AGAT1*) (**D**), later stage (*Vimentin*) (**E**) and mature cells (*Lamin B*) (**F**) in control and *Djubc9* RNAi animals. Statistical differences are measured by Student’s *t*-test and error bars indicate s.e.m. * *p* < 0.05, *** *p* < 0.001 (Student’s *t* test). For each time point, n = 15 with three experimental replicates. Scale bars: 200 µm.

**Figure 4 ijms-22-11707-f004:**
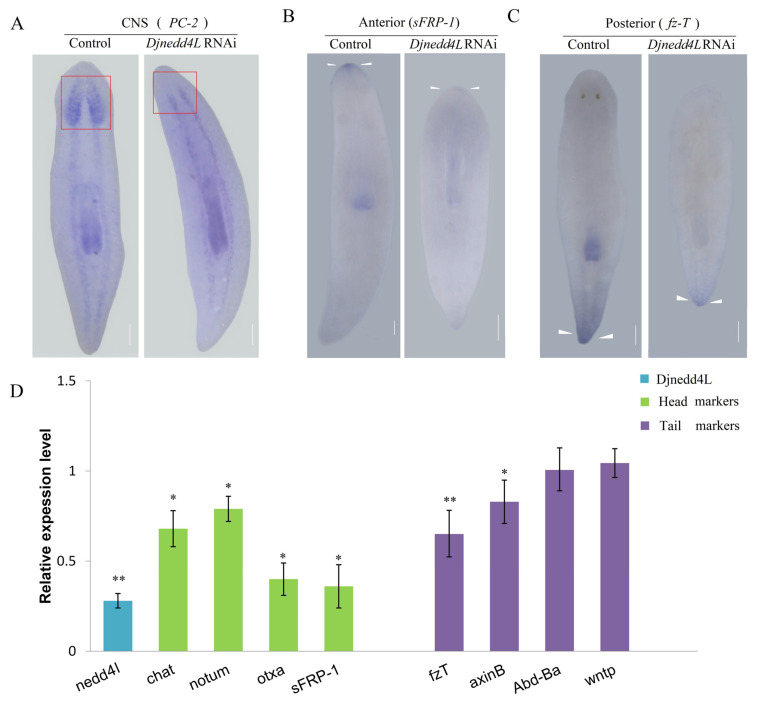
Analysis of the effect of *Djnedd4L* RNAi on the A-P polarity identities. (**A**–**C**) Characterization by in situ detection of CNS marker *PC-2*, anterior marker *sFRP-1*, and posterior marker *fz-T*. The cephalic brains are in red-shaded boxes (**A**). The arrows indicate the expression patterns of *sFRP-1* (**B**) and the normal expression patterns of *fz-T* (**C**). (**D**) Relative gene expression levels of head and tail markers in *Djnedd4L* RNAi animals. Error bars represent SDs of three biological replicates. Data were analyzed by Student’s *t*-test. * *p* < 0.05; ** *p* < 0.01; differences are considered significant at *p* < 0.05. Scale bars: 200 µm.

**Figure 5 ijms-22-11707-f005:**
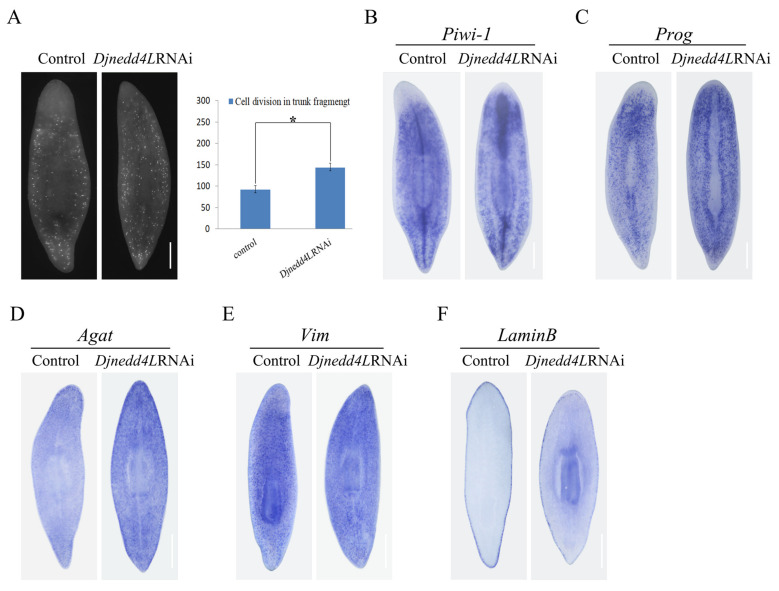
Analysis of cell division and stem cell lineage in the *Djnedd4L* (RNAi) phenotype. (**A**) Immunostaining with anti-phospho-histone 3 of trunk fragments. Worms were amputated pre- and post-pharyngeally and fixed at 8dpa. (**B**–**D**) Analysis of stem cells (*piwi-1*), early progeny (*prog*), and late progeny (*AGAT*) markers by whole-mount in situ hybridization (WISH) in regenerating trunk fragments in RNAi animals. (**E**,**F**) examination of epidermal cell (*vim*) and mature epidermal cells (lamin B). Error bars indicate s.e.m. n > 20 with at least three experimental replicates. * *p* < 0.05. Scale bars: 300 μm.

## Data Availability

All the data obtained in the current study are presented in this article. The RNA-Seq sequence raw data have been deposited at the National Center for Biotechnology Information (NCBI), Sequence Read Archive (SRA), under the Accession Number SAMN21193302.

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
