# Peer review of "Djnedd4L Is Required for Head Regeneration by Regulating Stem Cell Maintenance in Planarians"

_ijms, 2021, doi:10.3390/ijms222111707_

Round 1

Reviewer 1 Report

In the article entitled “Djnedd4L Is Required for Head Regeneration by Regulating the Stem Cell-Maintenance in Planarians” the authors investigated Planarians as a group of flatworms. Some planarian species have remarkable regenerative abilities, which involve abundant pluripotent adult stem cells. This makes these worms a powerful model system for understanding the molecular and evolutionary underpinnings of regeneration. By providing a succinct overview of planarian taxonomy, anatomy, available tools and the molecular orchestration of regeneration, this publication aims to showcase both the unique assets and the questions that can be addressed with this model system.

Thus, the objective of this publication is to assess Sumoylation and ubiquitylation as a homologous processes catalyzed by homologous enzymes. They are involved in nearly all aspects of eukaryotic biology. Planarians, which have the remarkable ability to regenerate their CNS (central nervous system), provide an excellent opportunity to investigate the molecular processes of CNS regeneration in vivo. In this study, the authors analyzed gene expression profiles during head regeneration with an RNA-seq-based screening approach and found that Djnedd4L and Djubc9 were required for head regeneration in planarians.

Planarians have long been known to possess astonishing regenerative capabilities. As succinctly stated by John Graham Dalyell in 1814, planarians ‘…may almost be called immortal under the edge of the knife’ (Dalyell, 1814). For example, if a planarian worm is chopped into three pieces, each of the piece’s regenerates back into a complete and perfectly proportioned animal within ∼2 weeks. In case of the tail (bottom) piece, this entails de novo formation of a head complete with brain, eyes and functional neuronal connections to the pre-existing tissue. Likewise, regeneration of the head (top) piece necessitates the de novo specification and formation of the trunk and tail. The central trunk (middle) piece needs to regenerate both a head and a tail; the fact that these always form at the front and rear of the piece, respectively, indicates that the regeneration process is primed by the polarity of pre-existing tissues.

The authors showed RNA interference targeting of Djubc9 caused the Phospho-H3 mitotic cells to decrease or even become absent as a part of the Djubc9 RNAi phenotype, which also showed the stem cell lineage collapsed along with the reduced expression of epidermal differentiation markers. Further, they found that Djnedd4L RNAi induced increased cell division and promote the premature differentiation during regeneration.

These results are in full agreement with the Core elements of the new head hypothesis. The ‘new head’ hypothesis proposed that the complexity and elaboration of the vertebrate head is a consequence of the advent of the migratory cranial neural crest and cranial placodes. Almost all placodes can be classified into two groups, namely (a) the neurogenic placodes and (b) the olfactory, lens, and otic placodes that contribute to sensory organs as L. Sommer described 2013 in “Patterning and Cell Type Specification in the Developing CNS and PNS”. The trigeminal and epibranchial (encompassing the geniculate, petrosal, and nodose) placodes belong to the first group and are neurogenic patches that contribute to the distal parts of the trigeminal ganglion (Vth) as well as to the distal ganglia of cranial nerves VII, IX, and X (McCabe and Bronner-Fraser, 2009; Streit, 2007). In contrast, the proximal parts of these ganglia and the glial cells of cranial ganglia derive from the neural crest. While trigeminal ganglia pass on somatosensory information from the head, the ganglia derived from epibranchial placodes transmit information from visceral organs and from the oral cavity. These new cell types enabled assembly of the craniofacial skeleton and a novel sensory system, which in turn allowed expansion of the anterior neuroepithelium to form the vertebrate brain. Morphological characters that arise from the neural crest and cranial placodes also allowed for transition from a predominantly filter ­feeding lifestyle of invertebrate chordates to the active predation.

During development, cranial neural crest cells migrate from the neural tube to populate the forming head. For example, the three epibranchial placodes (geniculate, petrosal and nodose) give rise to the seventh (VII), ninth (IX) and tenth (X) cranial nerves.

In their landmark 1983 paper, Gans and Northcutt proposed that the evolution of the "new head" of vertebrates was made possible by the formation of the neural crest and cranial bones.

The neural crest is a stem cell population that arises near the forming CNS and contributes to the formation of important cell types, including components of the peripheral nervous system and craniofacial skeleton, as well as elements of the cardiovascular system.

In recent years, the "new head" hypothesis has been challenged by the discovery of cells with some, but not all, features of vertebrate neural crest cells in invertebrate chordates.

Publications by the reviewer discuss recent findings on how neural crest cells may have evolved during vertebrate evolution. The results suggest that the repertoire of neural crest cells gradually expanded to include additional cell types during vertebrate evolution. These oral-derived neural crest-derived stem cells include those derived from germinal tissues of impacted wisdom teeth. These tissues are dental follicle cells (DFCs) and stem cells from the human dental apical papilla (SCAP). Furthermore, stem cells from the dental pulp of deciduous teeth (stem cells from human exfoliated decidious, SHED), permanent teeth (dental pulp stem cells, DPSCs) and from the periodontal ligament (periodontal ligament stem cells, PDLSCs) can be isolated. 

Taken together, the findings of the authors show that Djubc9 and Djnedd4L required for the stem cell maintenance in the planarian Dugesia japonica, which helps to elucidate the role of sumoylation and ubiquitylation in regulating the regeneration process.

“Minor issues

The results of this publication revealed, that Djubc9 and Djnedd4L required for the stem cell maintenance in the planarian Dugesia japonica. Agree, the regenerative powers of planarians derive largely from an abundant population of unusual adult stem cells, the neoblasts. Neoblasts are relatively small, round cells (7-12 µm in diameter) with a high nuclear-cytoplasmic volume ratio that are distributed throughout the planarian mesenchyme. Neoblasts are also crucial for the maintenance of planarian anatomy in the absence of wounding. Continual neoblast divisions and their resulting progeny continuously replace all differentiated cell types.

But above all, there is the question of how to generate the right cells at the right time and place, or, more specifically, how to guide differentiating progenitors through the maze of the planarian cell lineage tree. The reviewer would like to see a somewhat detailed discussion on the fact, that Planarians also offer an additional experimental approach to the shape and size challenge because of their general lack of a fixed body size. Planarians grow when fed and literally shrink when starving due to dynamic and food supply-dependent adjustments of total organismal cell numbers (Baguñà et al., 1990).

Author Response

As suggested by the reviewer, we have further discussed our findings on the model system of planarians. Planarians are an excellent model system for experimental dissection of processes such as tissue regeneration and tissue patterning. Planarian regeneration requires neoblasts to produces new cells of blastemas, which generate the replacement parts needed in regeneration. The fate specification in planarian regeneration primarily occurs in neoblasts, which are the source of new cells in planarian regeneration to produce all adult cell types. The capacity of neoblast for renewal and differentiation suggest that the neoblast population contains stem cells. During the processes of homeostasis and regeneration, stem cells must not only continually maintain the rates of cell division, but also induce differentiated progeny into new tissues properly. Our findings show that sumoylation and ubiquitylation is required for regeneration by regulating the maintenance of stem cell in the planarians Dugesia japonic, which helps to elucidate the role of sumoylation and ubiquitylation in regulating the regeneration process. Thank you for your suggestions.

Reviewer 2 Report

The Aauthors improved the quality of presented data and clarified important issues in the manuscript. I have bo additional comments.

Author Response

We are thankful for the referees to help us to improve our manuscript.